# Ethylene Signaling Modulates Dehydrin Expression in *Arabidopsis thaliana* Under Prolonged Dehydration

**DOI:** 10.3390/ijms26094148

**Published:** 2025-04-27

**Authors:** Irina I. Vaseva, Heorhii Balzhyk, Maria Trailova, Tsvetina Nikolova, Zornitsa Katerova, Simona Galabova, Dessislava Todorova, Iskren Sergiev, Valya Vassileva

**Affiliations:** Institute of Plant Physiology and Genetics, Bulgarian Academy of Sciences, Acad. Georgi Bonchev Street, Bldg. 21, 1113 Sofia, Bulgaria; h.balzhyk@gmail.com (H.B.); mimss_stoicheva@abv.bg (M.T.); tnikolova00@bio21.bas.bg (T.N.); zornitsa@bio21.bas.bg (Z.K.); sgalabova@bio21.bas.bg (S.G.); dessita@bio21.bas.bg (D.T.); iskren@bio21.bas.bg (I.S.); valyavassileva@bio21.bas.bg (V.V.)

**Keywords:** *Arabidopsis thaliana*, *ctr1-1* mutant, dehydrins, dehydration stress, ethylene signaling, *ein3eil1* mutant, ERF binding sites

## Abstract

Dehydrins are stress-inducible proteins with protective functions, characterized by high hydrophilicity, thermostability, and a low degree of secondary structure. They stabilize cellular membranes, preserve macromolecule conformation, and support enzymatic and structural protein functions. Their accumulation in plant tissues under drought is regulated by abscisic acid (ABA)-dependent and ABA-independent pathways. Ethylene plays a key role in stress adaptation, but its relationship with dehydrin accumulation remains unclear. This study investigates how ethylene influences dehydrin expression in *Arabidopsis thaliana* during prolonged dehydration using transcript profiling and immunodetection in wild-type (Col-0), ethylene-constitutive (*ctr1-1*), and ethylene-insensitive (*ein3eil1*) mutants. Comparative analyses showed increased survival of *ctr1-1* plants under dehydration stress, likely due to reduced oxidative damage. Analysis of dehydrin-coding genes identified multiple Ethylene Response Factor (ERF) binding sites, flanking the transcription start sites, which suggests a fine-tuned ethylene-dependent regulation. The ability of ethylene signaling to either suppress or stabilize particular dehydrins was demonstrated by RT-qPCR and immunodetection experiments. Under drought stress, ethylene signaling appeared to suppress root-specific dehydrins. A Y-segment-containing protein with approximate molecular weight of 20 kDa showed decreased levels in *ctr1-1* and higher accumulation in *ein3eil1*, indicating that ethylene signaling acts as a negative regulator. These results provide new information on the dual role of ethylene in dehydrin control, highlighting its function as a molecular switch in stress adaptive responses.

## 1. Introduction

Dehydration stress responses are regulated by the activation or suppression of multiple genes driven by abscisic acid (ABA)-dependent and ABA-independent signaling pathways [1,2,3,4]. Increased ABA levels under drought are directly linked to its regulatory function over the stomatal aperture to prevent water loss and the activation of a number of stress response mechanisms, which are often interrelated with other signaling pathways. For example, osmotic stress signaling is strongly influenced by the interaction between ABA and cytokinins, where reduced cytokinin signaling under suboptimal conditions causes growth inhibition [3]. Further, ethylene signaling together with gibberellins exert the primarily control of cell growth and proliferation in developing leaves under mild osmotic stress [5]. To gain deeper understanding of the molecular processes that govern plant tolerance and development under osmotic stress, the role of other plant hormones acting in parallel with ABA should be elucidated.

In this regard, the role of ethylene in the physiological response to dehydration remains incompletely understood. There are experimental data reporting both increased and decreased ethylene production in response to drought stress, as reviewed in [6]. In an earlier study conducted with wheat leaves, the authors found that ethylene production decreased as the severity of drought stress increased [7]. Another study performed in soybean reported that drought upregulated the enzymes involved in the synthesis of the ethylene precursor 1-aminocyclopropane-1-carboxylate (ACC), namely ACC synthase (ACS) and ACC oxidase (ACO), whereas one of the major negative regulators of ethylene signaling, CTR1, was downregulated, leading to higher levels of ethylene [8]. An increase in the synthesis of ACC deaminase in response to dehydration has also been reported [9]. This enzyme deactivates ACC, the precursor of ethylene, which reduces ethylene levels in the roots. This organ-specific physiological reaction counteracts the growth-inhibiting effects of ethylene, which is particularly important for root adaptation to limited soil moisture.

The capability of the gaseous hormone ethylene to either stimulate or inhibit the closing and opening of the stomata adds another layer of complexity to the aforementioned, seemingly contradictory research data. Ethylene-mediated stomatal closure has been shown to depend on the accumulation of hydrogen peroxide (H_2_O_2_) in leaves [10], which resembles the ABA-induced regulation. Meanwhile, some studies have indicated that elevated ABA concentrations suppress ethylene production [11,12]. Consequently, the dramatic increase in ABA concentration during prolonged water stress is expected to reduce ethylene production, as previously observed by Narayana et al. [7]. Ethylene inhibition of the ABA-mediated stomata closure has also been documented [13,14]. This negative crosstalk between the two stress hormones could be related to the ethylene interactions with auxins and cytokinins, which are known regulators of stomatal aperture under stress conditions [15].

Dehydration not only triggers stomatal closure but also strongly suppresses photosynthesis and cell growth, processes in which ethylene signaling plays a central role. The hormonal regulation of organ growth and stress responses is highly context-dependent, as evidenced by the variable results on endogenous ethylene levels and their link to drought stress adaptation. These findings outline the involvement of ethylene in stress signaling and adaptive growth. Arabidopsis rosette size has been shown to negatively correlate with drought stress resistance [16], suggesting potential functional implications of ethylene signaling in the improvement of drought stress resilience.Ethylene exerts its growth-inhibitory effects through interplay with local auxin synthesis and transport, mainly in epidermal cells [17]. A recent study on moisture-responsive root growth has identified ethylene as one of the main participants that control hydropatterning and root adaptation to water availability [18]. In this context, ethylene signaling seems to be an important element in the establishment of physiologically relevant molecular adjustments under dehydration [19,20].

Organ-specific dynamic changes in hormonal levels, including ethylene, are most likely linked to the physiological and biochemical modifications required for stress adaptation, such as the synthesis of stress-inducible metabolites and proteins that help the plant to withstand water limitation. Among these proteins, dehydrins comprise a family of stress-inducible molecules with protective functions that are synthesized in response to drought. They also accumulate in plant organs under high salinity and extreme temperature conditions [21]. Dehydrins play a crucial role in stabilizing cellular structures during stress [22]. They bind to membranes, sequester metal ions and reactive oxygen species (ROS), and retain water molecules, thereby protecting enzymes and proteins from oxidative damage caused by water stress. Additionally, some dehydrins translocate to the nucleus, where they form homodimers that safeguard DNA and transcription factors (TFs) involved in gene regulation [23].

Dehydrins are highly hydrophilic, thermostable proteins with a low degree of secondary and tertiary structure. Notably, they accumulate in particularly high concentrations during the later stages of ontogenesis, where they play a critical role in seed maturation, which explain their classification as Group II Late Embryogenesis Abundant (LEA) proteins. Dehydrins are rich in polar amino acids, including alanine (Ala), glycine (Gly), and serine (Ser), while containing relatively few hydrophobic residues. Their high hydrophilicity is mainly attributed to the lysine-rich K-segment (EKKGIMDKIKEKLPG), a conserved feature in all dehydrin types. Dehydrins have also a tyrosine-rich Y-segment (T/VDEYGNP) near the N-terminus and a serine-rich S-segment, which contains between four and ten serine residues [21]. Phosphorylation of the S-segment is believed to facilitate interactions with specific signaling peptides, enabling the translocation of dehydrins to the nucleus [23]. Additionally, phosphorylation of the S-segment has been linked to the regulation of dehydrin subcellular localization in response to stress conditions [24,25]. Recently, another 11-amino acid domain (DRGLFDFLGKK) has been identified as a conserved dehydrin feature, designated as the F-segment due to the presence of two hydrophobic phenylalanine (F) residues [26]. There are 10 annotated dehydrin genes in *Arabidopsis thaliana* genome (TAIR database, https://www.arabidopsis.org, accessed on 16 February 2025): *XERO1* (SK2), *XERO2* (K5), *COR47* (FSK3), *ERD10* (FSK3), *ERD14* (FSK2), *LEA* (Y3SK2), *RAB18* (Y2SK2), *HIRD11* (KS), *DHN AT4G38410* (FSK2) and *DHN-like AT4G39130* (K). Due to their unique molecular properties, these intrinsically disordered proteins have been found to stabilize the cell plasma membrane and maintain the native conformation of essential macromolecules. By the prevention of structural degradation, dehydrins ensure the proper functioning of enzymes, structural proteins and nucleic acids under unfavorable environmental conditions [27]. Plants subjected to water deficit and low-temperature stress express cytoplasmic dehydrins in various tissues. The dehydrin XERO2 improves cold resistance by binding to cell membranes, where the K-segment interacts with phospholipids through electrostatic interactions. This interaction restricts the mobility of the lipid phase and stabilizes associated membrane proteins. Many other members of the dehydrin family function similarly and are able to form aggregates to protect the integrity of the cell membrane. For example, the interaction of dehydrins COR47, ERD10, and RAB18 with membrane aquaporin AtPIP2B protects it from damage under abiotic stress [28]. A number of studies have reported a substantial increase in the expression of specific dehydrins associated with enhanced drought resistance. However, some dehydrin types have been found to be downregulated at mild or severe drought conditions [reviewed in 22], which reveals the complexity of dehydrin regulation and their diverse responses to drought depending on the stress severity, developmental stage, and affected organs. In silico analyses have shown that *A. thaliana* dehydrins accumulate predominantly in mature seeds (Appendix A), where they confer dehydration tolerance to the embryo—an essential function during this phase of the plant life cycle. *ERD14*, *XERO2,* and *HIRD11* genes are highly expressed in organs of plants grown under normal conditions, which suggests possible involvement of the respective proteins in fundamental growth and developmental processes (Appendix A).

Dehydrin synthesis is regulated by a complex network of molecular signals. While the stress hormone abscisic acid is well known for its role in controlling their expression, participation of other phytohormones in dehydrin regulation remains insufficiently explored. Although dehydrins are known to respond to wounding [29,30], in which the gaseous hormone plays an important role, studies on ethylene signaling in dehydrin gene expression are limited. However, existing evidence suggests that ethylene inhibits the C-repeat Binding Factors (CBFs)/Dehydration-responsive Element-binding 1 (DREB1) (CBF/DREB1) pathway, which controls the expression of cold-responsive dehydrins [31]. This inhibition is likely mediated by the action of Ethylene Insensitve 3 (EIN3) transcription factor (TF), a master regulator of ethylene responsive genes, which point to a potential regulatory interplay between ethylene signaling and dehydrin expression.

In *Arabidopsis*, ethylene is perceived by a family of transmembrane receptors, Ethylene Response 1 and 2 (ETR1, ETR2), Ethylene Response Sensor 1 and 2 (ERS1, ERS2), and Ethylene Insensitive 4 (EIN4), all of which reside in the endoplasmic reticulum membrane, reviewed in [32]. In the absence of ethylene, the signaling cascade is suppressed by Constitutive Triple Response 1 (CTR1), a Raf kinase that directly phosphorylates and inhibits the membrane protein Ethylene Insensitive 2 (EIN2). Ethylene binding to the receptors inactivates CTR1, releasing the repression of EIN2. This allows the C-terminal domain of EIN2 to be cleaved and translocated to the nucleus, where it activates the master ethylene TFs EIN3 and EIN3-Like 1 (EIL1). In the absence of ethylene, EIN3 is continuously degraded through a process controlled by the F-box proteins EIN3 Binding F-Box Protein 1 and 2 (EBF1 and EBF2). However, EIN2-C can regulate EIN3 stability by targeting *EBF1/2* mRNA to cytoplasmic processing-bodies (P-bodies), thereby preventing EIN3 degradation and sustaining ethylene signaling [33,34]. The aforementioned route is the linear canonical ethylene signaling pathway that is highly conserved in different plant species. Recent findings have indicated that Arabidopsis plants with enhanced nuclear-localized CTR1 exhibit improved tolerance to drought and salinity stress, suggesting a broader role for ethylene signaling in abiotic stress adaptation [35].

The present study evaluates the transcript abundance and accumulation of different dehydrin types in the leaves and roots of wild-type *A. thaliana* (Col-0), as well as ethylene constitutive (*ctr1-1*) and insensitive (*ein3eil1*) mutants subjected to prolonged dehydration induced by sorbitol. The central objective is to examine the role of ethylene signaling in the regulation of dehydrin expression. Addressing this largely understudied side of ethylene action will contribute to a better understanding of the complex regulatory mechanisms underlying dehydrin accumulation during water scarcity. Such knowledge could support the development of molecular strategies to create crop varieties with improved drought tolerance through the stable accumulation of physiologically relevant amounts of dehydrins in plant vegetative tissues.

## 2. Results

### 2.1. Survival and Accumulation of Stress Markers in the Wild-Type Col-0 and the Ethylene Signaling Mutants Subjected to Prolonged Dehydration

To assess the impact of ethylene signaling on dehydration stress response, we compared the survival rate and the physiological status of plants grown on ^1^/_2_ MS medium with or without 75 mM sorbitol for 21 days.

#### 2.1.1. Survival Rate

The survival rate of the three genotypes (Col-0, *ctr1-1* and *ein3eil1*) was assessed on nutrient medium containing 75 mM sorbitol for a period of 21 days (Figure 1a). During the initial stages of the test period, the survival rates of the individuals from the three genotypes did not differ significantly. A negative effect of sorbitol on the ethylene insensitive mutant *ein3eil1* was registered on Day 14 of the experimental period, and the survival rate dropped below 60% on the last day of measurements (Day 21). The wild-type and *ctr1-1* plants demonstrated similar survival rates which were more pronounced in the ethylene constitutive mutant, indicating its better predisposition to cope with the imposed dehydration stress (Figure 1a). The observed trend is consistent with previous studies, where constitutive ethylene signaling was associated with better resistance to abiotic stress compared to wild-type or ethylene-insensitive lines [36,37].

#### 2.1.2. L-Proline, MDA and H_2_O_2_ Content Measured in Col-0, *ctr1-1,* and *ein3eil1* After 21 Days of Growth on Sorbitol-Containing Medium

Under abiotic stress, plants initiate a cascade of synchronized responses to establish a first line of defense. This activation triggers transcriptional reprogramming to upregulate stress-responsive TFs and the accumulation of dehydrins and compatible solutes, such as free L-Proline (L-Pro), in the cytoplasm. In wild-type plants grown on sorbitol-containing medium, L-Pro levels were substantially elevated, reaching nearly 15-fold higher concentrations compared to the controls—1.8 µmol/g fresh weight (FW) in the controls vs. 29.6 µmol/g FW in the stressed group (Figure 1b).

The constitutive ethylene mutant *ctr1-1* showed a less pronounced increase in free L-Pro under dehydration, reaching approximately five-fold-higher levels compared to control conditions (1.7 µmol/g FW in the controls vs. 10.2 µmol/g FW in the stressed group). The double mutant *ein3eil1* exhibited elevated basal levels of this stress marker (4.9 µmol/g FW), approximately two times higher than those observed in Col-0 and *ctr1-1* under normal conditions (Figure 1b). Prolonged dehydration induced the accumulation of free L-Pro in the leaves of *ein3eil1* to levels comparable to those observed in stressed wild-type plants (32.9 µmol/g FW).

Cell membrane damage was evaluated by the accumulation of malondialdehyde (MDA). Wild-type plants (Col-0) showed the highest increase in MDA levels under dehydration with approximately a two-fold rise compared to their respective controls (25 nmol/g FW in the controls vs. 55 nmol/g FW in the stressed group). In the absence of stress, *ein3eil1* already showed elevated MDA levels (47.9 nmol/g FW), nearly double those in Col-0 and *ctr1-1* (Figure 1c). Upon dehydration, MDA content in *ein3eil1* increased by a further 60%, reaching 77 nmol/g FW—the highest among the tested genotypes. The *ctr1-1* showed a milder increase with MDA levels rising by about 40% compared to its respective control group (Figure 1c).

Under stress, plant cell receptors detect signals from signaling molecules like Ca^2^⁺, nitric oxide (NO) and H_2_O_2_, which activate plant defense mechanisms. In addition to being a signaling molecule, H_2_O_2_ accumulation also serves as a conventional stress marker. In dehydrated wild-type plants, the estimated content of H_2_O_2_ was approximately 50% higher than in the controls (612.3 nmol/g FW in the controls vs. 971.5 nmol/g FW in the stressed group). The *ein3eil1* mutant had elevated basal H_2_O_2_ levels (780.5 nmol/g FW) compared to the controls of the other two genotypes (Col-0 and *ctr1-1*), which increased by around 30% upon dehydration (1033.1 nmol/g FW). The *ctr1-1* mutant again showed the least pronounced H_2_O_2_ increase, approximately 14% above control levels (Figure 1d).

Overall, the better survival rate of the *ctr1-1* constitutive ethylene signaling mutant correlated with lower levels of the measured stress markers compared to the wild-type and the ethylene-insensitive *ein3eil1* plants.

### 2.2. In Silico Analyses of Promoter Regions of the Dehydrin-Coding A. thaliana Genes

There is a close relationship between dehydrin expression patterns and the presence of *cis*-regulatory elements in their promoter regions [38]. Dehydration-responsive Element/C-repeat (DRE/CRT) and DRE-/CRT-binding protein 2 (DREB2) TFs play key roles in ABA-independent gene expression under osmotic stress [1]. Most promoters of dehydrin-coding genes contain multiple binding sites for DREB TFs (Appendix A), which belong to the Apetala2/Ethylene Responsive Factors (AP2/ERF) TF family. ERFs represent a separate subgroup in the AP2/ERF superfamily. According to the Eukaryotic Promoter Database (EPD, https://epd.expasy.org/epd (accessed on 11 May 2024)), binding sites for 27 different ERFs are available in the promoter regions of the analyzed *A. thaliana* dehydrin genes (Figure 2). The analysis revealed the presence of ERF binding sites downstream of the Transcription Start Site (TSS) in many of the dehydrin-encoding genes, particularly abundant in the coding regions of *LEA*, *XERO1*, *RAB18*, *DHN AT4G38410,* and *DHN-like AT4G39130*. Their positions are highlighted in pink in Figure 2. *ERD14* and *XERO2* are the only dehydrin-coding genes that do not exhibit this feature. The functional significance of TF binding sites located downstream of the TSS has recently gained attention in a publication by Voichek et al. [39], which demonstrated that such sites can modulate transcription in a dose-dependent manner, acting much like a rheostat to fine-tune gene expression across different cell types in vascular plants. Based on the observed distribution of ERF binding sites upstream and downstream of the TSS in *A. thaliana* dehydrin genes, a similar regulatory mechanism may also be implicated here.

Additional confirmation for the involvement of ethylene signals in the regulation of dehydrin genes was obtained from an experiment with plants grown on medium supplemented with ACC (Appendix A). The ethylene precursor strongly inhibited the expression of most of the *A. thaliana* dehydrin-coding genes.

The performed in silico analyses of the promoter regions of *A. thalina* dehydrin genes confirmed the presence of multiple ERF binding sites. Some of them were localized downstream of the TSS, a pattern previously associated with the dose-dependent regulation of gene expression in other systems.

### 2.3. Transcript Profiling of Dehydrin-Coding Genes in Organs of A. thaliana Wild-Type (Col-0) and Ethylene Mutants (ctr1-1 and ein3eil1) Under Prolonged Dehydration

Organ-specific transcript profiling of the 10 annotated Arabidopsis dehydrin genes was performed by RT-qPCR in Col-0, *ctr1-1,* and *ein3eil1* plants grown for 21 days on ^1^/_2_ MS medium supplemented with 75 mM sorbitol.

#### 2.3.1. Dehydrin Transcript Profiles in Leaves

Gene expression analyses of leaf samples showed that prolonged dehydration led to a moderate increase in *XERO1* expression (coding for a SK2-type dehydrin) in wild-type Col-0 plants. The expression levels of other dehydrin genes (*COR47*, *XERO2*, *RAB18*, *LEA*, *ERD10*) remained similar to their respective controls or were slightly inhibited (*ERD14*, *HIRD11*, *DHN AT4G38410*, and *DHN-like AT4G391*) (Figure 3).

Transcript levels of several dehydrin genes (*COR47*, *XERO2*, *RAB18*, *LEA*, *ERD10*, *ERD14*, *DHN AT4G38410,* and *DHN-like AT4G39130*) were increased in the leaves of the constitutive mutant *ctr1-1* compared to those in the controls. *XERO1* appeared to be less affected by dehydration, while *HIRD11* was the only dehydrin transcript downregulated in *ctr1-1* leaf samples (Figure 3). The ethylene-insensitive double mutant *ein3eil1* exhibited distinct differences in leaf dehydrin expression patterns with downregulated *LEA*, *ERD10,* and *ERD14*, and upregulated *XERO1* (as observed in the wild-type dehydrated leaf samples) and *XERO2* genes. Expression levels of *COR47*, *RAB18*, *DHN AT4G38410,* and *DHN-like AT4G391* remained comparable to those in *ein3eil1* control plants (Figure 3). Significant differences in relative expression levels in the leaf samples among the three genotypes were observed for *XERO1*, *LEA*, *ERD10*, *ERD14,* and *DHN AT4G38410*, suggesting an involvement of ethylene signaling in their regulation.

The SK2-type dehydrin-coding gene *XERO1* had altered expression in both ethylene mutants compared to the wild type. *DHN AT4G38410* was downregulated in wild-type leaves following prolonged sorbitol exposure, but its transcript levels in the ethylene mutants remained less affected by the stress. *LEA*, a Y3SK2 dehydrin-coding gene, showed slightly elevated (above the control level) expression in the leaves of sorbitol-grown *ctr1-1* plants but was downregulated in *ein3eil1.* Similar changes were observed for *ERD10* (FSK3-type protein) and *ERD14* (FSK2 type protein), which exhibited slightly increased transcript levels in *ctr1-1*, but suppressed expression in *ein3eil1* (Figure 3). The consistent downregulation of *HIRD11* in wild-type and both ethylene mutants subjected to dehydration, together with the limited number of ERF binding sites identified in its promoter region, suggest that ethylene signaling does not perform a central role in the control of its expression (Figure 3).

Overall, under prolonged dehydration stress, the disruption of ethylene signaling downregulated *XERO1*, which was the only transcript with elevated levels in the stressed wild-type plants. This effect diminished in both samples derived from *ctr1-1* and *ein3eil1* stressed plants. A more stable expression of the other dehydrin genes was observed in the ethylene constitutive mutant. The double mutant *ein3eil1* accumulated lower amounts of *LEA*, *ERD10* and *ERD14* transcripts in the leaves compared to stressed wild-type and *ctr1-1* plants.

#### 2.3.2. Dehydrin Transcript Profiles in Roots

Transcript profiling in roots revealed that *RAB18* (coding a Y2SK2-type dehydrin) was the only gene significantly upregulated in Col-0, showing an 8–12-fold increase after 21 days of growth on dehydration-inducing medium. The expression of the same gene in both ethylene mutants (*ctr1-1* and *ein3eil1*) was only slightly altered with levels close to those observed in their respective controls (Figure 4). Expression levels of the other monitored dehydrin-coding genes fluctuated around or slightly above the control baseline. The only gene consistently downregulated in response to sorbitol was *ERD10* in the ethylene-insensitive mutant *ein3eil1*. However, no statistically significant differences in dehydrin expression were detected among the wild-type and the ethylene mutant genotypes.

### 2.4. Immunodetection of Dehydrin Types in Leaves and Roots of A. thaliana Wild Type (Col-0) and Ethylene Mutants (ctr1-1 and ein3eil1) Under Prolonged Dehydration

Immunodetection of dehydrin proteins was performed on total soluble protein extracts from the leaves and roots of 21-day-old control and sorbitol-stressed plants. Equal amounts of soluble protein (10 µg) from each experimental group were separated via 10% SDS-PAGE and subsequently transferred onto nitrocellulose membranes. The membranes were probed with three different primary antibodies recognizing the conservative dehydrin segments K, Y, and S (Figure 5). To confirm equal protein loading across samples, the intact membranes were visualized by transient Ponceau S staining prior to immunoblotting (Figure 5 and Appendix A). As an internal reference, we developed equally loaded membranes and probed them with a monoclonal plant-specific actin antibody (Appendix A). The reference protein showed equal actin abundance in the leaf samples of the tested genotypes but varying levels in the root samples with stronger immunosignal in *ctr1-1* roots compared to Col-0- and *ein3eil1*-derived samples.

#### 2.4.1. Dehydrin Immunosignals in Leaves

Immunoblot analysis of dehydrin proteins extracted from the leaves of the three genotypes showed distinct expression patterns. The strongest immunosignals detected with the K-segment primary antibody were positioned between 45 kDa and 35 kDa, forming three well-defined bands with varying intensities among control and stressed samples in all three genotypes (Figure 5a). Weaker K-immunosignals were detected bellow 30 kDa, also exhibiting genotype-specific variations. In Col-0 leaves, the most pronounced relative increase (stressed vs. control) in sorbitol-exposed plants was registered for signals in the 15–40 kDa range. Band intensities at 45 kDa remained similar, suggesting a weaker impact of dehydration stress on this particular dehydrin signal (Figure 5a).The immunoreactive band migrating around 20 kDa position was revealed with both K- and Y-antibodies, suggesting that the applied stress increased the levels of a Y-segment containing dehydrin in Col-0 leaves. The only two annotated Y-segment-containing members of the Arabidopsis dehydrin family are dehydrin LEA (ID: Q96261, 19.3 kDa) and RAB18 (ID: P30185, 18.5 kDa). The Y immunosignal observed at ~40 kDa position was likely due to dimerization, as no Y-containing dehydrin of this molecular weight is known to exist in Arabidopsis.

Quantification of the K-segment immunosignals (Figure 5a) using ImageJ revealed that in *ctr1-1*, dehydrin band intensities remained unchanged (15–20 kDa and 30–35 kDa) or slightly decreased (45 kDa) in sorbitol-exposed samples compared to their respective controls. The only exception was the ~40 kDa K-immunoreactive band (marked in blue on the blot), which showed increased intensity in sorbitol-treated *ctr1-1* samples compared to untreated controls (Figure 5a). The quantification of the leaf *ctr1-1* Y-signal, migrating around 20 kDa position, revealed increased intensity in the stressed samples compared to the control mutant plants, but the rise was less pronounced than in the wild-type. The other Y-antibody-reactive band in the sorbitol-stressed leaves of *ctr1-1* (around 40 kDa position) showed 90% intensity of the respective *ctr1-1* control.

Surprisingly stable K immunosignals were detected in the double mutant samples. Even in the controls, their intensity was comparable to that detected in wild-type plants grown under sorbitol-induced stress (Figure 5a). This observation suggests that the attenuation of ethylene signaling caused by deficiencies in the EIN3 and EIL1 TFs led to an increased or mostly stabilized accumulation of certain dehydrin types in the leaves of *ein3eil1* (Figure 5a, see bar chart). Probing with a primary antibody against the Y-segment confirmed that bands at positions around 20 kDa and 40 kDa contained this conservative domain (Figure 5a). The immunoblot profiles suggest that ethylene signaling affected the accumulation of a Y-segment containing dehydrin with an approximate molecular weight (MW) of 20 kDa (Figure 5a).

It is important to note that the detected immunosignals do not always align with the predicted MW of dehydrin proteins. This discrepancy is largely attributed to their propensity for homo- and heterodimeric formation [40,41] or post-translational modifications, such as phosphorylation [23,24,25], which can alter their apparent molecular size during electrophoresis. Particularly, the S-segment is highly susceptible to phosphorylation under stress conditions [24,25], which might also prevent its recognition by the S-segment-specific primary antibody. This could explain the relatively weak detection of S-immunosignal in our experiments. The only S-immunoreactive band appeared at an approximate MW of 70 kDa. Since no *A. thalina* dehydrin of this size has been annotated, this signal was likely an artifact or may have originated from homo- or heterodimerization of S-containing dehydrins (Figure 5a).

In summary, the leaf dehydrin immunoprofiles revealed that the constitutive ethylene signaling was negatively correlated with the accumulation of a Y-segment-containing dehydrin monomer, which was apparent in the leaf *ctr1-1* samples.

#### 2.4.2. DHN Immunosignals in Roots

The abundance of dehydrin immunosignals in roots was substantially lower than in leaf samples, despite equal loading of soluble proteins on the membranes (Figure 5b). A well-defined band, aligning with the 60 kDa position of the protein marker, was detected by the K-segment primary antibody. Since no *A. thaliana* dehydrin of such MW has been annotated, this signal is likely a result of homo- or hetero-oligomerization. Bands of similar size were also detected by Y- and S-segment antibodies (Ab-Y and Ab-S), suggesting that the protein may belong to the YnSKn-type dehydrin family (Figure 5b).

Some weaker signals identified by Ab-Y and Ab-S were not visualized on the K-probed membrane, which could be due to detection limitations or sample-specific sensitivity of the conserved K-segment. Relative quantification of band intensities (compared to control samples of the same genotype) showed that upon dehydration, dehydrin levels decreased in wild-type roots but increased substantially in *ein3eil1* (Figure 5b, bar chart). A similar trend was observed for Ab-Y and Ab-S signals, reinforcing the idea that ethylene signaling contributes to the regulation of root dehydrin levels. The weak immunosignals detected in *ctr1-1* root samples with all three antibodies seemed largely unaffected by dehydration stress (Figure 5b).

The comparison between leaf and root immunosignal profiles outlined a possible organ-specific role of ethylene signaling in dehydrin accumulation.

## 3. Discussion

Drought is one of the most severe climate extremes, negatively affecting plant growth and productivity and causing serious ecological and economic consequences. The impacts of water stress depend on its duration and severity—some plants are able to recover and survive, while others experience irreversible damage. Understanding the precise mechanisms that plants have developed to withstand prolonged water limitation is essential for the development of novel molecular strategies to improve stress resilience. Drought-induced damage disrupts cellular homeostasis and leads to osmotic and oxidative stress. In response, plants synthesize signaling molecules and protective compounds, many of which also serve as diagnostic stress markers [42]. The improved survival rate of *ctr1-1* plants grown on sorbitol, corroborated well with the modest induction of oxidative stress markers (H_2_O_2_, MDA, and proline). This implies that ethylene signaling plays an important but not yet fully characterized role in adaptive drought responses. Recent research has added new information on the involvement of ethylene signaling in drought adaptation [43]. The authors reported that the ethylene-responsive TFs EIN3 and EIL1 are part of cascade mechanisms that induce H_2_O_2_ accumulation, which, in turn, acts as a signal for hairy roots’ initiation—an adaptive trait that increases root surface area for improved water uptake under drought conditions. The results from the present study provide new information on the lesser-understood role of ethylene in drought adaptation. Beyond its well-documented function in growth inhibition, ethylene also influences the accumulation of dehydrins under prolonged dehydration, further reinforcing its multifaceted participation in plant stress resistance.

Experimental evidence has demonstrated that DREBs and ERFs can function as positive and negative regulators of the expression of osmotic stress-responsive genes [1,44], which suggests that a similar dual regulatory mechanism may apply to dehydrin gene expression via these TFs. The *Arabidopsis* genome contains 122 ERF genes that are classified into several groups based on their specific molecular characteristics and functions [45]. These TFs regulate growth and development, and they play a key role in responses to various abiotic and biotic stresses, such as salinity, drought, pathogen attacks, etc. [45,46]. In this study, 27 different ERF binding sites were identified in the promoter regions of *A. thaliana* dehydrin genes with notable presence downstream of TSS in many of them. The recently reported ability of TSS-proximal binding sites to modulate gene expression in a dose-dependent manner in plants [39] suggests that ethylene signaling might act as a regulatory switch for dehydrin expression, depending on environmental conditions. EPD analysis revealed multiple ERF binding sites from groups I and V (such as ERF4, ERF13, and ERF15) in the promoters of dehydrin genes. These TFs regulate the response to biotic and abiotic stresses by either activating or repressing ABA-dependent genes [47,48]. In addition, binding sites for ERFs from group IXc, specifically ERF96 and ERF98, were also identified in dehydrin gene promoters. This is a relatively small group, consisting of only four members, each with a small molecular size (131–139 amino acids) [45]. ERF98 has been associated with the regulation of processes related to salt tolerance [49,50], while ERF96 is involved in ABA-mediated stress responses [45]. The binding sites for ERF96 were identified downstream of TSS in *LEA*, *XERO1*, *DHN At4g38410*, *DHN-like At4g39130,* and *RAB18.* EPD analyses also revealed that ERF98 binding sites downstream of TSS in *DHN At4g38410* and *DHN-like At4g39130.* ERF96 and ERF98 are expressed in almost all plant tissues and organs, except roots [45], which may have implications for the tissue-specific, ethylene-mediated regulation of dehydrin expression. Further, binding sites for ERF104 and ERF105 were identified in the TSS regions of *LEA*, *XERO1*, *COR47*, *DHN At4g38410,* and *DHN-like At4g39130*. Since these TFs are predominantly expressed in roots [51], it is likely that these ethylene-related TFs also exert organ-specific control over dehydrin gene expression.

Expression profiling of dehydrin genes under dehydration stress confirmed that ethylene signaling indeed plays a role in the regulation of dehydrin transcripts in an organ-specific way. The ethylene-insensitive *ein3eil1* mutant was characterized with downregulation of *LEA*, *ERD10,* and *ERD14* in leaves, while in roots, only *ERD10* transcripts were found to be reduced compared to the control. This indicates that ethylene may control *LEA* and *ERD14* expression in an organ-specific manner.

The relatively stable transcript profiles of many dehydrin genes in the organs of the ethylene-constitutive mutant *ctr1-1* is also consistent with the assumption that ethylene takes part in the regulation of dehydrin expression. Additionally, the observation that the ethylene precursor ACC inhibited the expression of most dehydrin-coding genes in *A. thaliana* further supports the role of ethylene in the modulation of dehydrin synthesis and accumulation. Meanwhile, the *HIRD11* gene was consistently downregulated across all genotypes, implying that its expression is likely less dependent on ethylene signaling.

The levels of *ERD10* transcripts, coding for a FSK3 protein with predicted MW of 29.5 kDa, measured in Col-0 leaf samples, corresponded to the intensity of the immunosignal migrating at a position around 30 kDa, which also appeared to be weaker in the two ethylene mutants, thus matching the RT-qPCR profiling of the respective gene. However, it is well established that transcript levels do not always correlate with the measured levels of the respective protein. Therefore, the correlation between RT-qPCR data and immunodetection profiles should be interpreted with caution, keeping in mind that discrepancies between gene expression and protein abundance, as measured by techniques like Western blotting, can arise from various factors affecting transcriptional regulation, post-transcriptional processing, translation efficiency, and protein stability.

Western blot analyses frequently revealed MW shifts in dehydrins, providing evidence of their ability to form homodimers, heterodimers, or interactions with other macromolecules. In the present study, some of the detected bands on the immunoblots did not match the predicted MWs of Arabidopsis dehydrins, likely due to homo-/heterodimerization or post-translational modifications. Consequently, the correlation between transcript profiling, particularly for genes encoding YnSKn-type dehydrins, and the immunodetection results with primary antibodies against Y- and S-segment, is more difficult to interpret. The sole S-segment signal (~70 kDa) detected in the leaf samples could originate from protein clustering, as the only two annotated YnSKn dehydrin-coding genes in Arabidopsis are *LEA* (encoding a Y3SK2 protein with a predicted MW of 19.3 kDa) and *RAB18* (encoding a Y2SK2 protein with a predicted MW of 18.5 kDa). A study by Hébert-Haché et al. [52] identified six dehydrin-related protein bands in grapevine buds using an antibody against the conserved K-segment. Their results confirmed the presence of dehydrins at 23, 41, 48, and 90 kDa, with the 90 kDa band likely representing higher-order oligomerization. Similarly, the Y- and S-immunosignals with a higher than predicted MW, detected in the present study, could reflect probable YnSKn dehydrin oligomerization.

The Y-segment antibody also confirmed the presence of a Y-motif-containing dehydrin in leaves (approximate size of 20 kDa), which was negatively affected by ethylene signaling, as evidenced by the lower intensity of the immunoreactive band in *ctr1-1* samples and the stronger signal detected in *ein3eil1* samples.

Dehydrin accumulation was substantially lower in roots than in leaves, despite equal protein loading of the nitrocellulose membranes used for immunodetection. A prominent 60 kDa band, detected with the K-segment antibody, suggests that the signal likely originated from dehydrin clustering or oligomerization, rather than from a single annotated Arabidopsis dehydrin. Immunodetection with Ab-Y and Ab-S further confirmed the presence of a probable YnSKn-type dehydrin, although the relatively weaker signals observed with these antibodies may indicate limitations or variations in detection sensitivity or antibody specificity. Relative quantification of the immunosignals showed that dehydrin levels decreased by 50% in wild-type roots under dehydration but increased significantly in the ethylene-insensitive *ein3eil1* mutant, suggesting that ethylene signaling may act as a suppressor of certain dehydrin isoforms under stress conditions. In contrast, *ctr1-1* roots showed weak dehydrin immunosignals that remained largely unaffected by dehydration, indicating that constitutive ethylene signaling stabilizes dehydrin levels. These findings highlight the dual role of ethylene in both suppressing and stabilizing different dehydrins, particularly in plant roots. Further investigation, preferably with organ- and cell type-specific resolution, is needed to determine whether this regulatory effect is mediated at the transcriptional level through ethylene-responsive TFs or post-translationally via effects on dehydrin stability and modifications.

## 4. Materials and Methods

### 4.1. Plant Material and Growth Conditions

Wild-type *Arabidopsis thaliana* (Col-0), the ethylene constitutive mutant (*ctr1-1*), and the ethylene-insensitive double mutant *ein3eil1*, which has attenuated expression of two major ethylene TFs, EIN3 and EIL1, were used in this study. Seeds were obtained from the Nottingham Arabidopsis Stock Center (NASC, http://arabidopsis.info (accessed on 10 October 2022)) and the Arabidopsis Biological Resource Center (ABRC), Ohio State University (https://abrc.osu.edu (accessed on 10 October 2022)). All mutants are developed in Col-0 genetic background. Seeds were surface sterilized and stratified for 2 days at 4 °C in darkness before germination. Plants were then grown for 21 days (21 DAG) on solid half-strength Murashige and Skoog (^1^/_2_ MS) medium (pH 5.7). Control plants were grown on standard ^1^/_2_ MS medium, whereas dehydration stress was induced by supplementing the medium with 75 mM sorbitol [53]. Plates were kept in a growth chamber under controlled conditions: 50% relative air humidity, a temperature of 22 °C, a 16 h light/8 h dark photoperiod, and a light intensity of approximately 150 μmol m^−2^ s^−1^.

### 4.2. Survival Test

Plants were grown on medium supplemented with 75 mM sorbitol, and the number of survived individuals from each tested genotype was recorded on 4, 7, 14, and 21 DAG.

### 4.3. Measurements of MDA, H_2_O_2_, and Free Proline Contents

The levels of stress markers (MDA, H_2_O_2_, and free L-Pro) in the rosettes of survived individuals were measured at 21 DAG. Plant material (250 mg) was homogenized in 4 mL 1% (*w*/*v*) cold trichloroacetic acid and centrifuged at 15,000× *g* at 4 °C for 30 min. The resulting supernatant was for the analysis of MDA, H_2_O_2_ and proline contents. MDA was quantified as thiobarbituric acid-reagent product using an extinction coefficient (ε) of 155 mM^−1^ cm^−1^ [54]. Hydrogen peroxide was assessed using the iodide method, following a modified protocol from Frew et al. [55]. A 75 µL aliquot of supernatant was incubated with 75 µL of 1 M KI for 1 h at room temperature in the dark, and absorbance was read at 352 nm. The concentration of H_2_O_2_ was calculated using a molar extinction coefficient of 23 mM^−1^ cm^−1^. Free proline content was determined following the method described by Bates et al. [56], using a standard curve for quantification.

### 4.4. RT-qPCR Analysis of A. thaliana Dehydrin Genes

Total RNA was extracted from leaves and roots using the RNeasy Mini Kit (QIAGEN, Venlo, The Netherlands), according to the procedure described by the manufacturer. Two hundred and fifty nanograms (250 ng) of total RNA were used to synthesize cDNA with the FIREScript^®^ RT cDNA synthesis MIX (Solis BioDyne, Tartu, Estonia), following the manufacturer’s protocol. Dehydrin transcript profiling was performed by quantitative real-time RT-PCR (RT-qPCR) with HOT FIREPol^®^ EvaGreen^®^ qPCR Supermix (Solis BioDyne, Tartu, Estonia) on a PikoReal Real-Time PCR System (Thermo Scientific, Basel, Switzerland).

The qPCR protocol consisted of the following steps: initial denaturation at 95 °C for 12 min; 45 cycles of 95 °C for 15 s (denaturation); 60 °C for 20 s (annealing); and 72 °C for 20 s (extension). The final step comprised a melting curve analysis of the PCR products to verify that the detected fluorescence comes from a single amplicon (60 °C–95 °C temperature range, 0.2 °C increments for 60 s). Gene expression was normalized using the reference gene *EF-1ɑ* (*At5g60390*), and relative expression levels were calculated according to the ΔΔCq method [57]. The primer pairs used in the RT-qPCR analyses are listed in Table 1.

### 4.5. Immunodetection of Dehydrins

Soluble proteins were extracted from leaf (200 mg) and root (100 mg) samples of 21-day-old plants grown on ½ MS medium −/+ 75 mM sorbitol, using ice-cold 100 mM Tris–HCl buffer (pH 7.4). Equal amounts of total soluble protein (10 µg per lane) were separated by 10% SDS-PAGE and transferred to a nitrocellulose membrane using the Trans-Blot system (Bio-Rad, Hercules, CA, USA). ROTI^®^-Mark WESTERN PLUS (Carl Roth GmbH, Karlsruhe, Germany) was used as a MW reference. Equal loading and the quality of the protein transfer were verified by staining the membrane with Ponceau S (Sigma-Aldrich, Darmstadt, Germany). After destaining with TBS buffer (0.1 M Tris, pH 7.9, 0.15 M NaCl), the membranes were blocked overnight at 4 °C with 5% milk (Carl Roth GmbH, Karlsruhe, Germany). Immunodetection with primary antibodies against the dehydrin K-, S-, and Y-segments followed a previously described procedure [58]. A monoclonal plant-specific actin antibody (Sigma, Saint Louis, MO, USA) was used to detect the actin levels in the leaves and root samples.

Membranes, probed with anti K-, anti Y-, and anti S-Abs, were incubated for 2 h at room temperature with an anti-rabbit IgG horseradish peroxidase-conjugated secondary antibody (Agrisera, Umeå, Sweden). The reference immunoblots developed with the anti-actin Ab were incubated for 2 h at room temperature with anti-mouse IgG horseradish peroxidase-conjugated secondary antibody (Agrisera, Umeå, Sweden). The secondary antibodies were diluted in TBS-T buffer (TBS supplemented with 0.01% Tween 20), and the incubation of the membranes was performed under continuous agitation.

After washing with TBS-T, membranes loaded with leaf and root samples were developed for 5 min with reagent ECL Bright and ECL SuperBright (Agrisera, Umeå, Sweden), respectively. Immunosignals were visualized with G:BOX Mini 9 Gel Documentation System (Syngene, Cambridge, UK), and signal intensities were quantified with ImageJ 1.52 r.

### 4.6. Statistical Analysis

At least 20 individuals per genotype were used in the experiments, which were repeated three times. Measurements of stress markers and RT-qPCRs analyses were conducted in triplicates in three independent experiments. Data were evaluated with R-4.5.0 (Figure 1) and Python 3.12.0 (Figure 2 and Figure 3) [59,60]. The line chart in Figure 1a represents the average values with standard deviation (SD). The box plots in Figure 1b–d, and Figure 2 and Figure 3 show the highest, median, and the lowest value of three datasets. One-way ANOVA followed by Tukey’s HSD post-hoc range test was used to assess statistically significant differences at *p* ≤0.05.

## 5. Conclusions

The results indicate that *ctr1-1*, which is characterized by constitutive ethylene signaling, is better adapted to dehydration stress, likely due to the role of ethylene in mitigating oxidative damage, as reflected by the lower levels of stress markers in the mutant. The findings also suggest that ethylene contributes to the complex regulation of dehydrin expression, as evidenced by the presence of ERF binding sites upstream and downstream of the TSS in 8 out of the 10 annotated Arabidopsis dehydrin genes. This observation points to a potential ethylene-related regulatory mechanism for fine-tuning their expression. Further support for this hypothesis comes from RT-qPCR and immunodetection data, which demonstrated that ethylene signaling could either suppress or stabilize specific dehydrins. We found that the ethylene insensitive mutant *ein3eil1* maintained higher levels of certain dehydrins compared to *ctr1-1*. The immunodetection of a Y-segment-containing dehydrin with approximate MW of 20 kDa evidenced that it was particularly sensitive to ethylene, exhibiting reduced levels in *ctr1-1* and increased accumulation in *ein3eil1*. Although RT-qPCR data and immunoblot analyses of leaf samples are difficult to interpret, the results obtained from root samples suggest that this effect may be associated with the RAB18 dehydrin. Overall, this study highlights the complex and dual role of ethylene in modulating dehydrin accumulation under dehydration stress, providing new information on its regulatory function in plant stress adaptation. In summary, we found that:ethylene signaling fine-tuned the expression of specific dehydrin genes;ethylene signaling was negatively correlated with the accumulation of Y-segment-containing dehydrins in Arabidopsis.

## Figures and Tables

**Figure 1 ijms-26-04148-f001:**
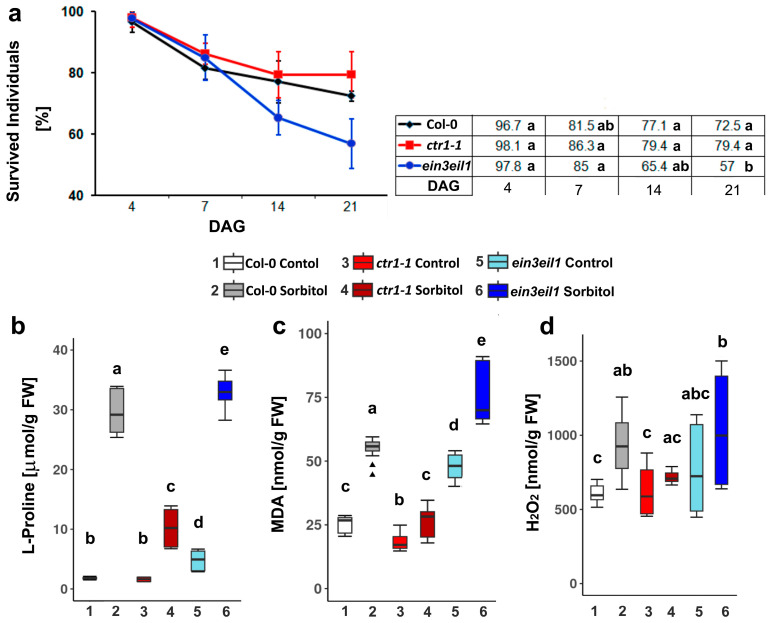
Percentage of survived individuals (error bars represent standard deviation, n ≥ 20) (**a**), and levels of stress markers: free L-Proline (**b**), malondialdehyde (MDA) (**c**), and hydrogen peroxide (**d**) accumulated in Col-0, *ctr1-1*,and *ein3eil1* plants grown on ½ MS −/+ 75 mM sorbitol for 21 days after germination (DAG) (different letters indicate statistically significant differences at *p* < 0.05, one-way ANOVA with Tukey’s HSD post-hoc test; the outliers in the box plots are indicated with triangles).

**Figure 2 ijms-26-04148-f002:**
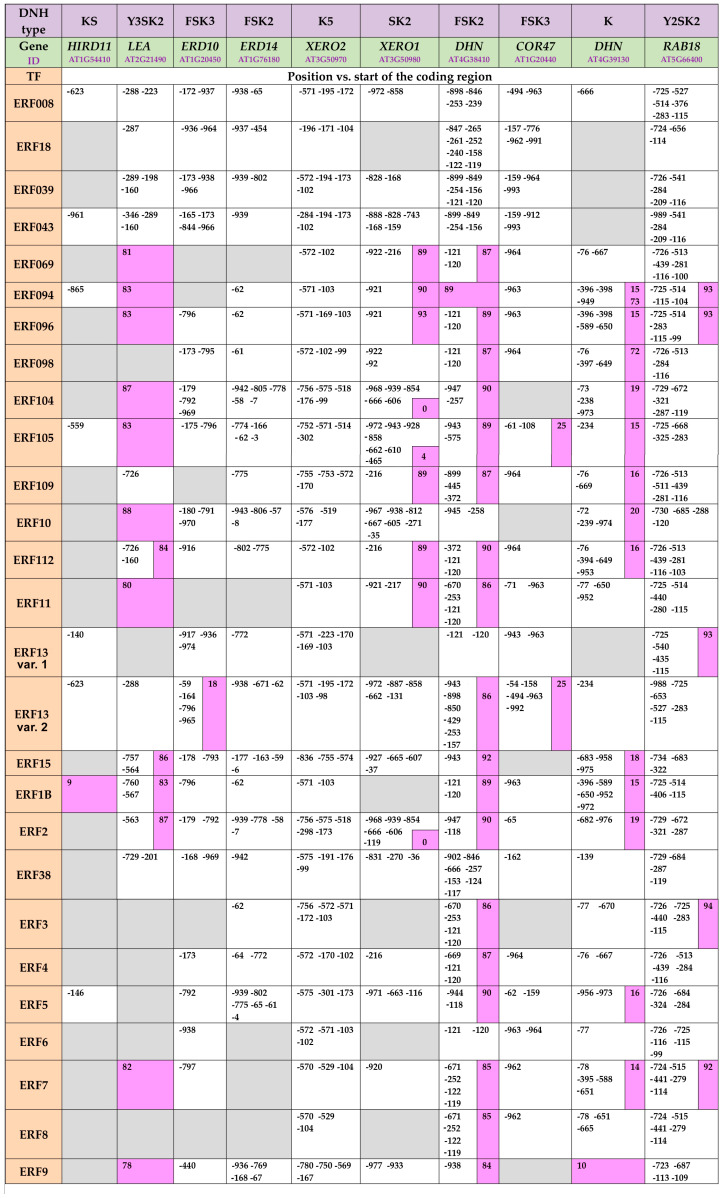
Positions of Ethylene Response Factors (ERF)-binding sites in the promoter regions of *Arabidopsis thaliana* dehydrin-coding genes according to Eukaryotic Promoter Database, (https://epd.expasy.org/epd/, accessed on 11 May 2024). The grey color indicates the lack of sites for a specific ERF, and the pink color marks binding sites located downstream of TSS.

**Figure 3 ijms-26-04148-f003:**
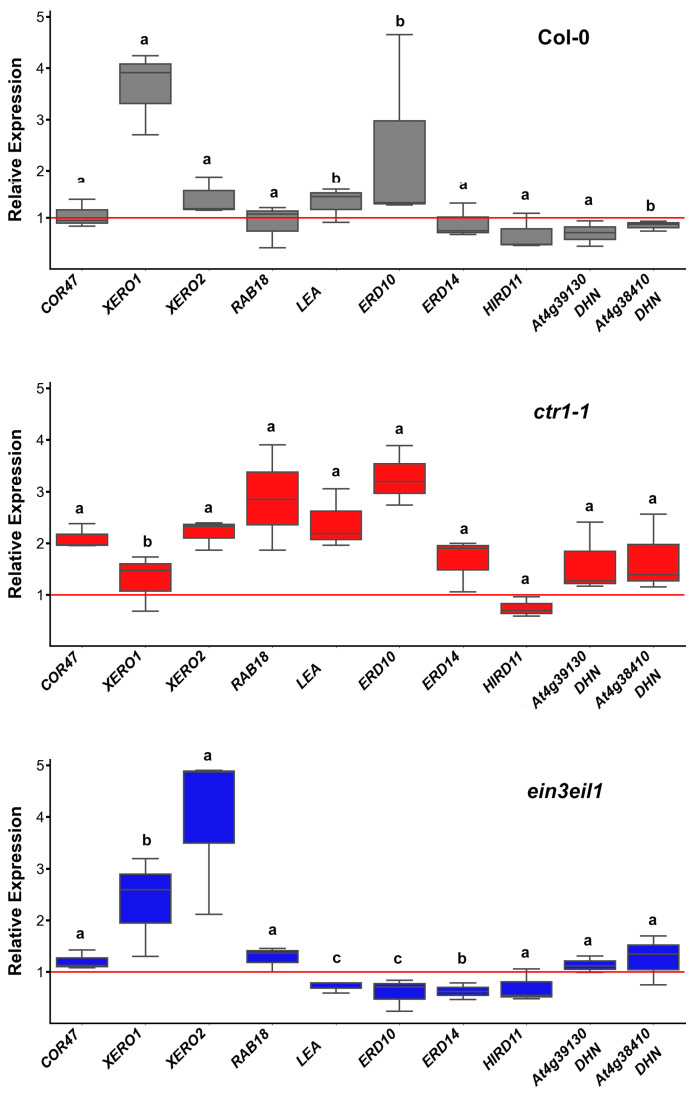
Relative expression of dehydrin genes in the leaves of wild-type (Col-0), the constitutive ethylene response mutant (*ctr1-1*) and the ethylene-insensitive double mutant (*ein3eil1*) grown on ^1^/_2_ MS medium −/+ 75 mM sorbitol. The plots represent data from three independent experiments, depicting the median (horizontal line in each box), the lowest and the highest values (whiskers endpoints). The red line indicates the basal expression level of each gene measured in control samples of the same genotype. Different letters indicate statistically significant differences in gene expression among the three genotypes (*p* < 0.05, one-way ANOVA).

**Figure 4 ijms-26-04148-f004:**
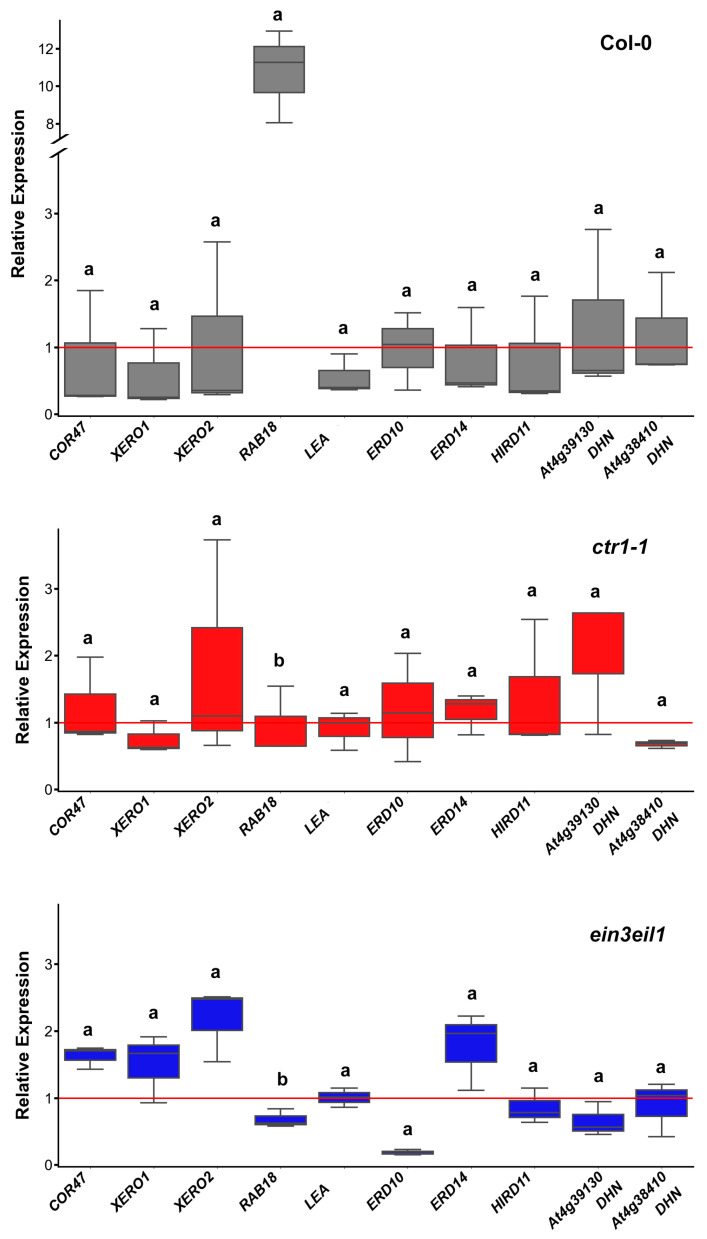
Relative expression of dehydrin genes in the roots of wild type (Col-0), the constitutive ethylene response mutant (*ctr1-1*) and the ethylene-insensitive double mutant (*ein3eil1*) grown on ^1^/_2_ MS medium −/+ 75 mM sorbitol. The plots represent data from three independent experiments, depicting the median (horizontal line in each box), the lowest and the highest values (whisker endpoints). The red line indicates the basal expression level of each gene measured in control samples of the same genotype. Different letters designate statistically significant differences in gene expression among the three genotypes (*p* < 0.05, one-way ANOVA).

**Figure 5 ijms-26-04148-f005:**
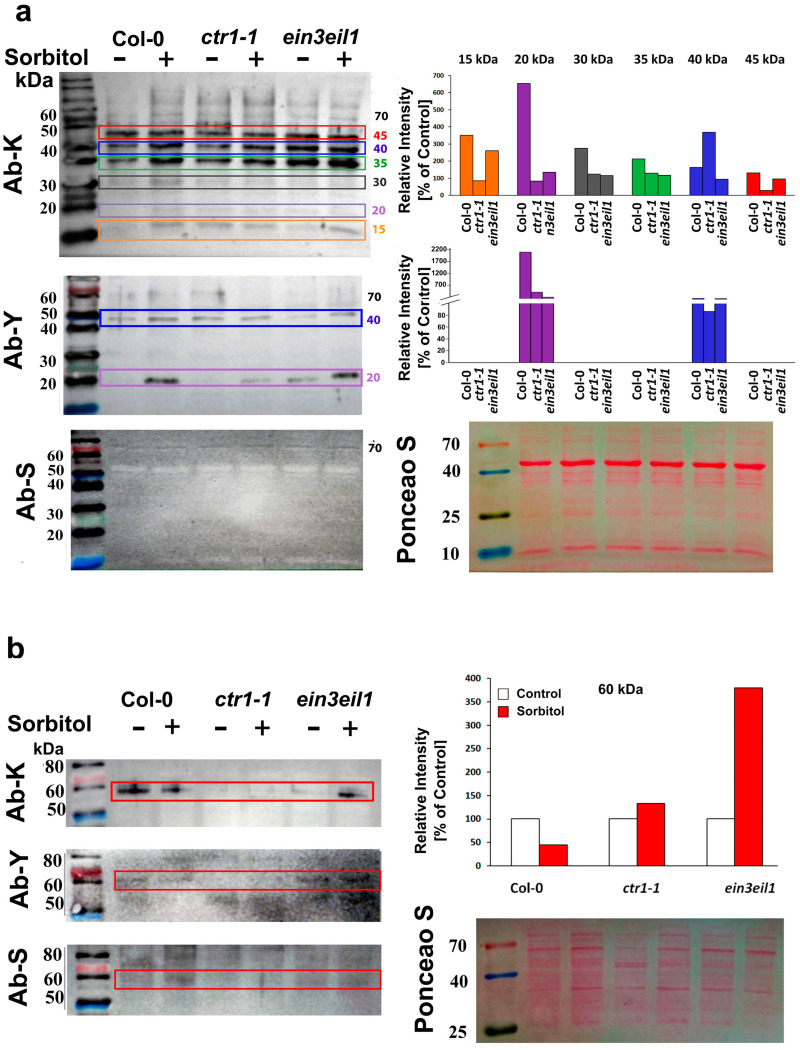
Immunodetection of dehydrin proteins in leaves (**a**) and roots (**b**) of Col-0, *ctr1-1* and *ein3eil1* plants grown for 21 days on ^1^/_2_ MS medium −/+ 75 mM sorbitol. Representative images of membranes probed with primary antibodies against the K-, Y-, and S-segments, and Ponceau S staining of total protein loads are shown. The intensities of K- and Y-segment immunoreactive bands in leaves (aligned with the 15, 20, 30, 35, 40, and 45 kDa positions of the protein marker), and K-signal in root samples (at the 60 kDa position), are expressed as values relative to the respective control (quantified with ImageJ 1.52 r).

**Table 1 ijms-26-04148-t001:** Primer pairs used in the RT-qPCR analyses.

Gene Name	Locus	Forward Primer (5′-3′)	Reverse Primer (5′-3′)
*HIRD11*	*At1g54410*	cacgacggagaaggcaaaag	gatgcacggttctcctgtct
*LEA*	*At2g21490*	ccggtgttgttagctccact	agatgctcttcaagcgaccc
*ERD10*	*At1g20450*	ggggaaacacttgtttactcaatgg	tggcggaacggactaatttca
*ERD14*	*At1g76180*	gaacaggaggtgccaaaggt	aagaactgtcgcttcggtga
*XERO1*	*At3g50980*	gggtcatcacgactccaaca	ccatgcaacgaccataagcg
*XERO2*	*At3g50970*	caaactgggactaacacggc	ctagtgatgaccaccgggaag
*RAB18*	*At5g66400*	caccacgccgacattttctg	cggttctggtaagacgccat
*COR47*	*At1g20440*	ctacgacggagcttccagtg	cgaatgtcccactcccacat
*DHN*	*At4g38410*	cattggtcgacgaaacgcag	cgtttagccaaggtgttgcc
*DHN*	*At4g39130*	cgagtttggtaacgccatgc	ggtcttgaagagggtcgtgg
*EF-1* *ɑ*	*At5g60390*	cagatcggcaacggctac	gagaaggtctccaccaccat

## Data Availability

Research data are available upon request from the corresponding author.

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
