# Peer review of "Ethylene Signaling Modulates Dehydrin Expression in Arabidopsis thaliana Under Prolonged Dehydration"

_ijms, 2025, doi:10.3390/ijms26094148_

Round 1

Reviewer 1 Report

Comments and Suggestions for Authors

The manuscript investigates the role of ethylene signaling in modulating dehydrin expression during prolonged dehydration in Arabidopsis thaliana. While the study addresses an important topic and provides some novel insights, the current version has several limitations that require revision to strengthen the scientific rigor and clarity. Below are my comments for improvement.

Major Comments:

  1. Selection of Dehydrin Genes for qPCR Analysis

While the authors selected 10 annotated dehydrin genes for RT-qPCR analysis and supported this with preliminary bioinformatics data (e.g., promoter analysis of ERF-binding sites), the rationale for focusing exclusively on these genes remains unclear. A more systematic approach would involve genome-wide transcript profiling (e.g., RNA-seq) under dehydration stress to identify differentially expressed genes (DEGs), followed by targeted validation of dehydrins. This would strengthen the justification for selecting specific dehydrins and reduce potential bias. If RNA-seq data are unavailable, citing prior studies demonstrating dehydrin-specific responses to ethylene signaling could contextualize the gene selection.

  1. Western Blot Methodology and Quantification

(1). Antibody Specificity vs. Gene Expression: The immunoblot analysis used antibodies targeting conserved dehydrin domains (K-, Y-, and S-segments) rather than isoform-specific antibodies. While this approach is valid for detecting dehydrin family members, it creates a disconnect between the RT-qPCR data (gene-specific) and the protein-level results (family-wide). For instance, the 20 kDa Y-segment-containing protein cannot be directly linked to any of the 10 genes analyzed by RT-qPCR.

(2). Lack of Loading Controls: The absence of internal reference proteins (e.g., β-actin, tubulin, or total protein staining) in Western blots undermines quantitative comparisons. For example, the decreased 45 kDa K-segment signal in ctr1-1 (Fig. 4a) could reflect either true protein downregulation or unequal loading.

Minor Comments:

  1. Formatting of Gene and Plant Latin Symbols

Please ensure that all gene names and Latin plant species names are italicized throughout the manuscript, including those in the references (Reference 41). Additionally, all protein names should remain in plain text to maintain proper scientific formatting.

  1. Consistency in Reference Formatting

Some reference titles have inconsistencies in capitalization—some words are capitalized while others are not. Please carefully review and standardize the formatting so that all reference titles follow a consistent capitalization style, in accordance with the journal's guidelines.

Comments on the Quality of English Language

The overall quality of the English language in this article is generally well-written, yet there are certain aspects that necessitate refinement.

  1. In lines 182-183, “acumulation” should be corrected to “accumulation” , “ethylen” should be corrected to “ethylene” , “wild type” should be corrected to “wild-type”
  2. In line 386, “relatively weak S-immunosignal detection” should be corrected to “relatively weak detection of S-immunosignal”
  3. In line 470, “relatvely” should be corrected to “relatively”

Author Response

Comment 1: “The manuscript investigates the role of ethylene signaling in modulating dehydrin expression during prolonged dehydration in Arabidopsis thaliana. While the study addresses an important topic and provides some novel insights, the current version has several limitations that require revision to strengthen the scientific rigor and clarity. Below are my comments for improvement.

Major Comments:

  1. Selection of Dehydrin Genes for qPCR Analysis

While the authors selected 10 annotated dehydrin genes for RT-qPCR analysis and supported this with preliminary bioinformatics data (e.g., promoter analysis of ERF-binding sites), the rationale for focusing exclusively on these genes remains unclear. A more systematic approach would involve genome-wide transcript profiling (e.g., RNA-seq) under dehydration stress to identify differentially expressed genes (DEGs), followed by targeted validation of dehydrins. This would strengthen the justification for selecting specific dehydrins and reduce potential bias.

Response 1: We would like to thank Reviewer 1 for the comment. Indeed, RNA-seq under dehydration stress, followed by targeted validation of DEGs, including dehydrins, is part of our broader research plan related to the evaluation of the ethylene signaling in drought stress response. However, the current manuscript has a different focus: as clearly stated in the last paragraph of the Introduction, our primary objective was to examine the role of ethylene signaling in the regulation of dehydrin expression under prolonged dehydration. Therefore, we include in the RT-qPCR analysis the full list of all annotated A. thaliana DHN genes in TAIR database. The chosen experimental approach aimed to avoid selection bias and provide an extensive assessment of the dehydrin family and how it relates to ethylene signaling. We used the respective mutants (ctr1-1 and the double mutant ein3eil1) to directly test the involvement of major signaling components (CTR1, EIN3 and EIL1) in the regulation of dehydrin expression. This strategy gives an informative view of the role of ethylene signaling in the expression and accumulation of these stress-inducible proteins with protective functions. Although ABA is the primary hormone known to regulate dehydrin genes, the involvement of other phytohormones, including ethylene, in the control of their accumulation under stress remains unclear. Our data suggest that ethylene can act as a modulator, either suppressing or stabilizing the expression of specific dehydrin genes in an organ-specific manner. To our knowledge, no prior studies have investigated this relationship, particularly for the dehydrin family, making this work a novel contribution to the field.

Comment 2: If RNA-seq data are unavailable, citing prior studies demonstrating dehydrin-specific responses to ethylene signaling could contextualize the gene selection.

Response 2: Published data linking ethylene signaling to the regulation of dehydrin genes are currently very limited or nonexistent. This knowledge gap was what motivated our study: to evaluate the potential involvement of the ethylene signaling pathway in dehydrin expression. We have addressed this lack of information in the Introduction (page 3, the last paragraph). The few studies that are “touching” the subject indirectly are included in the reference list (Reference 29: Richard et al. Plant Mol Biol 2000; 30: Mota et al. Front Plant Sci 2019; 31: Robison et al. Front Plant Sci 2019).

Comment 3: “2. Western Blot Methodology and Quantification

(1). Antibody Specificity vs. Gene Expression: The immunoblot analysis used antibodies targeting conserved dehydrin domains (K-, Y-, and S-segments) rather than isoform-specific antibodies. While this approach is valid for detecting dehydrin family members, it creates a disconnect between the RT-qPCR data (gene-specific) and the protein-level results (family-wide).”

Response 3: It is true that a direct comparison between gene-specific RT-qPCR data and protein-level immunodetection is complicated by the use of antibodies targeting conserved dehydrin domains rather than isoform-specific epitopes. However, isoform-specific antibodies typically recognize structural differences that arise from modifications at specific amino acids or conformational changes. Since dehydrins are intrinsically disordered proteins lacking defined secondary or tertiary structures, their molecular structure resembles that of flexible polypeptides, which make the recognition of isoform-specific antibody particularly challenging. Thus, such antibodies may not have the specificity required at the peptide level. Instead, the use of antibodies against the conserved K-, Y- and S-segments is predominantly considered in dehydrin immuno-screening, as they enable the detection of different dehydrin types based on their conserved motifs. Moreover, due to their unique molecular properties, these proteins are prone to posttranslational modifications (e.g.,  phosphorilation) and tend to form homo- or heterodimers, as already discussed in the manuscript (Discussion, page 14). These properties further complicate the interpretation of protein size and abundance on immunoblots, and make problematic to establish a direct correlation with gene-specific RT-qPCR data. However, the combined immunodetection using the three different Ab against the conservative DHN domains provides informative clues about the dehydrin types that show divergent profiles in the tested ethylene mutant genotypes.

In the revised variant of Figure 4 we have added a graph showing quantification of the detected leaf Y-immunosignals in leaves (panel a). It confirms the results obtained with the anti-K primary antibody in the leaf samples; more specifically, the dehydration-induced Y-signal in the wild-type (at position 20 kDa) is negatively affected by the disrupted ethylene signaling in ctr1-1 and ein3eil1

Comment 4: For instance, the 20 kDa Y-segment-containing protein cannot be directly linked to any of the 10 genes analyzed by RT-qPCR.

Response 4: The detected Y-segment-containing protein aligns approximately with the 20 kDa position of the protein marker. We have referred to its size using appropriately cautious phrasing, such as “positions around 20 kDa” and “approximate molecular weight (MW) of 20 kDa” to reflect the limitations of gel-based size estimation. 

Among the ten annotated Arabidopsis dehydrin genes analyzed by RT-qPCR, four encode proteins with predicted MW around 20 kDa: Dehydrin LEA (19.3 kDa); RAB18 (18.5 kDa), ERD14 (20.7 kDa) and XERO2 (20.9 kDa). Of these, only LEA and RAB18 contain Y-segments, and are therefore detectable by the anti-Y antibody used in our immunoblot assays.

Indeed, the RT-qPCR expression profiles of At2g21490 (Dehydrin LEA) and AT5g66400 (RAB18), do not directly correlate with the observed Y-immunosignals. Several factors may contribute to this discrepancy, including variations in transcript half-life, post-translational modifications, and the well-established tendency of dehydrins to form homo- or hetero-oligomers (as discussed in Hernández-Sánchez et al. 2017). These properties often result in altered migration patterns during SDS-PAGE with bands differing from expected monomeric sizes. In this context, the Y-immunosignal detected around 40 kDa is likely due to dimerization, as no monomeric Y-segment-containing dehydrin of this size (40 kDa) has been identified in Arabidopsis. Consequently, direct correlation between gene expression levels and immunodetected protein bands must be interpreted with caution, especially in the case of proteins like dehydrins that show dynamic structural behavior.

Comment 5: “(2). Lack of Loading Controls: The absence of internal reference proteins (e.g., β-actin, tubulin, or total protein staining) in Western blots undermines quantitative comparisons. For example, the decreased 45 kDa K-segment signal in ctr1-1 (Fig. 4a) could reflect either true protein downregulation or unequal loading.”

Response 5: In Figure S3 (Supplementary materials), we have provided images of the intact membranes along with the respective Ponceau S staining to demonstrate equal total protein loading, as stated in the Material and methods section (page 16). To address this comment more clearly, we have now included representative Ponceau S staining profiles directly in the main figures: Figure 4a (leaf samples) and Figure 4b (root samples). Further, we used a monoclonal plant-specific actin antibody (Sigma, Saint Louis, USA) as an internal protein reference. These control immunoblots were carried out on equally loaded membranes and are now included in Figure S3. The actin immunosignal showed equal abundance in leaf samples of the tested genotypes, supporting valid comparisons of the dehydrin immunosignals. In root samples, actin levels varied slightly, with stronger signals detected in ctr1-1 compared to Col-0 and ein3eil1. These additional reference immunoblots have been added to Figure S3 and mentioned also in Results (page 11) of the revised manuscript. We hope that the additional controls clarify the validity of the protein quantifications presented.

Minor Comments

“1. Formatting of Gene and Plant Latin Symbols

Please ensure that all gene names and Latin plant species names are italicized throughout the manuscript, including those in the references (Reference 41). Additionally, all protein names should remain in plain text to maintain proper scientific formatting.

Response: The text has been carefully revised, and all formatting inconsistencies regarding gene names, Latin species names and protein symbols have been corrected, according to the reviewer’s comment.

“2. Consistency in Reference Formatting”

Some reference titles have inconsistencies in capitalization—some words are capitalized while others are not. Please carefully review and standardize the formatting so that all reference titles follow a consistent capitalization style, in accordance with the journal's guidelines.

Response: We have edited the reference list following the formatting requirements of IJMS.

Comments on the Quality of English Language

The overall quality of the English language in this article is generally well-written, yet there are certain aspects that necessitate refinement.

  1. In lines 182-183, “acumulation” should be corrected to “accumulation” , “ethylen” should be corrected to “ethylene” , “wild type” should be corrected to “wild-type”
  2. In line 386, “relatively weak S-immunosignal detection” should be corrected to “relatively weak detection of S-immunosignal”
  3. In line 470, “relatvely” should be corrected to “relatively”

Response: The text has been carefully checked for Englisg style and grammar mistakes. The identified typos have been corrected.

Reviewer 2 Report

Comments and Suggestions for Authors

The authors in this manuscript studied the role of ethylene in dehydrin gene expression under dought stress in Arabidopsis. They performed a survival analysis that showed that ctr1 constitutive ethylene signaling mutants had better survival than Wt and ein3eil1 ethylene insensitive mutants. ctr1 mutant also had less accumulation of stress markers like H2O2.

They also perfomed in silico analysis of dehydrin promoter to find TF binding elements.

The authors utilized expression studies of the dehydrin genes under drought stress using qRT-PCR and immunobloting. The expression studies showed that ethylene signaling modulates some of the dehydrin gene expression.

Some general and precific comments on the manuscript:

The introduction is sufficient and well presented.

Material and methods adequately describe the experiments performed, an exception to that is the source for the antibodies used in the immunoblotting.

The results should have a small conclusion of what did the experiment showed that the scientific community didn't know up until now.

Table 1, could be transfered to the supplementary

Fig.2 and Fig.3 could benefit from a more traditional bar plot presentation to see the each genotype before and after the dought stress for each gene.

The biggest weakness of the manuscript is the limited data provided. The experiments show some involvement of ethylene signaling in dehydrin expression but not clear pattern. Some gene with larger differences in expression are not studied further.

The immunoblotting data are confusing, do the antibodies bind to all the dehydrins with the specific domain? in that case they only provide a general view of the dehydrin expression that is not very helpful. GFP tagged lines or custom antibodies for the dehydrin genes could provide more useful information.

Author Response

Comment 1:The authors in this manuscript studied the role of ethylene in dehydrin gene expression under dought stress in Arabidopsis. They performed a survival analysis that showed that ctr1 constitutive ethylene signaling mutants had better survival than Wt and ein3eil1 ethylene insensitive mutants. ctr1 mutant also had less accumulation of stress markers like H2O2.

They also perfomed in silico analysis of dehydrin promoter to find TF binding elements.

The authors utilized expression studies of the dehydrin genes under drought stress using qRT-PCR and immunobloting. The expression studies showed that ethylene signaling modulates some of the dehydrin gene expression.

Some general and precific comments on the manuscript:

The introduction is sufficient and well presented.

Material and methods adequately describe the experiments performed, an exception to that is the source for the antibodies used in the immunoblotting.

Response 1: We would like to thank the reviewer for the positive evaluation of the Introduction and the Materials and Methods sections. Regarding the source of the antibodies, they were kindly shared by Prof. Urs Feller from Bern University. His contribution is acknowledged in the Acknowledgments section of the manuscript.

Comment 2:The results should have a small conclusion of what did the experiment showed that the scientific community didn't know up until now.

Response 2: We have added a brief concluding remarks at the end of each subsection of Results to underline the novel findings and their contribution to current knowledge, as recommended.

Comment 3:Table 1, could be transfered to the supplementary

Response 3: We appreciate the suggestion of the reviewer; however, we believe that Table 1 provides essential information that supports key points discussed in the main text. For this reason, we prefer to retain it in the main body of the manuscript, which ensures accessibility for the readers.

Comment 4:Fig.2 and Fig.3 could benefit from a more traditional bar plot presentation to see the each genotype before and after the dought stress for each gene.

Response 4: We agree that bar plots are a widely used and familiar format. In fact, we initially prepared several visualization formats for the RT-qPCR data, including the suggested traditional bar charts; however, we chose box-and-whisker plots as they provide more informative representations of data distribution, variability and outliers across the three independent biological replicates. The gene expression profiles were calculated using the 2−ΔΔCT method, where the expression level of the non-treated control samples is regarded as 1, which is depicted by the red line in the graphs. A conventional bar chart format would show identical bar heights for the control samples across genotypes, providing limited visual information. In contrast, the current format enables clearer comparison of the changes in gene expression levels and the variability among genotypes in response to drought stress.

Comment 5:The biggest weakness of the manuscript is the limited data provided. The experiments show some involvement of ethylene signaling in dehydrin expression but not clear pattern. Some gene with larger differences in expression are not studied further.

Response 5: We agree that the regulatory effects of ethylene signaling on dehydrin expression do not follow a simple, uniform pattern. This complexity reflects the broader nature of stress-responsive gene networks, where multiple hormones and signaling pathways often interact. To date, abscisic acid (ABA) has been the primary hormonal regulator associated with the expression of dehydrin genes, but the role of ethylene remains largely unexplored. The main objective of our study, as stated in the manuscript, was to assess whether ethylene signaling plays a role in the regulation of dehydrin accumulation during prolonged dehydration stress. We acknowledge the Reviewer’s critical comment but also would like to outline the complexity of the research effort made in the present study.  

We combined transcript-level analysis with immunodetection assays to assess  gene expression and protein accumulation in ethylene signaling mutants and wild-type plants. We recognize the limitations of this approach, particularly given that transcriptomic and proteomic data do not always correlate directly. This is especially true for dehydrins, which pose additional challenges due to their (1) the intrinsically disordered structure, lacking stable secondary and tertiary conformations; (2) susceptibility to posttranslational modifications; and (3) tendency to form homo- and hetero-oligomers as part of their functional state (as discussed in Hernández-Sánchez et al., 2017). These properties often lead to discrepancies between mRNA and protein-level data and complicate straightforward interpretation. We believe that our data provide a valuable starting point by the identification of ethylene signaling as a potential modulator of specific dehydrin genes and protein sub-classes. In particular, our findings suggest a negative correlation between ethylene signaling and the accumulation of Y-segment-containing dehydrins. We have expanded the relevant discussion in the revised manuscript to clarify this interpretation.

In our future work, it is planned to build upon these findings and include cell-type-specific analysis of ethylene-regulated YSK-type dehydrin expression to achieve more detailed resolution.

Finally, we have summarized the new findings in the Conclusion section to clearly articulate the contribution of this work:

  • ethylene signaling fine-tunes the expression of specific dehydrin genes;
  • ethylene signaling is negatively correlated with the accumulation of Y-segment-containing dehydrins in Arabidopsis. Data from RT-qPCR and immunoblot analyses of root samples suggest that this effect may be associated with RAB18; however, discrepancies in leaf samples indicate additional regulatory complexity.

We hope that the added clarification and future directions outlined here help to address the reviewer’s concerns.

Comment 6: The immunoblotting data are confusing, do the antibodies bind to all the dehydrins with the specific domain? in that case they only provide a general view of the dehydrin expression that is not very helpful. GFP tagged lines or custom antibodies for the dehydrin genes could provide more useful information.”

Response 6: We appreciate the Revieer’s comment and the suggestion for alternative approaches. Indeed, the recognition of epitopes by custom-made antibodies often depends on specific protein conformations. However, this presents a limitation in the case of dehydrins, which are intrinsically disordered. Their moleculesdo not adopt stable secondary or tertiary structures.  For this reason, development of isoform-specific antibodies is difficult and often unreliable for this protein family. Dehydrin immunodetection is routinely done by antibodies recognizing the conserved motif - the K-segment, which is present in all dehydrin proteins. All commercially available anti-DHN antibodies are based on this feature. However, only some dehydrins contain Y- or S-segments, allowing further subtype differentiation. In our study, we employed a combined immunodetection approach with anti-K, anti-Y and anti-S antibodies. This strategy allowed us to distinguish among different dehydrin subtypes (e.g., K-type vs. YSK-type), which provides a more detailed understanding of how ethylene signaling affects specific classes of dehydrins. In particular, it enabled us to observe differences in the accumulation of YSK-type dehydrins in the tested genotypes, which would not have been possible with a single general anti-DHN antibody. We agree that future studies employing GFP-tagged lines or epitope-tagged dehydrin variants could provide even more specific information for the regulation of individual genes and localization of proteins. However, given the current limitations in available resources and the structural nature of dehydrins, our approach represents a valid and informative method for the evaluation of ethylene-regulated dehydrin profiles.

Round 2

Reviewer 1 Report

Comments and Suggestions for Authors

The authors have addressed my previous concerns appropriately, and the manuscript has significantly improved in clarity and scientific rigor. I appreciate their efforts in revising the work. I have no further major concerns, and I believe the manuscript is now suitable for publication.